# Detection of Equus Caballus Papillomavirus Type-2 in Asymptomatic Italian Horses

**DOI:** 10.3390/v14081696

**Published:** 2022-07-31

**Authors:** Katia Cappelli, Chiara Grazia De Ciucis, Samanta Mecocci, Tiziana Nervo, Maria Ines Crescio, Marco Pepe, Rodolfo Gialletti, Daniele Pietrucci, Laura Federica Migone, Silvia Turco, Luca Mechelli, Fabrizio Passamonti, Carlo Drago, Gian Guido Donato, Katia Varello, Paola Modesto, Giovanni Chillemi, Alessandro Ghelardi, Elisabetta Razzuoli

**Affiliations:** 1Dipartimento di Medicina Veterinaria, 06126 Perugia, Italy; katia.cappelli@unipg.it (K.C.); marco.pepe@unipg.it (M.P.); rodolfo.gialletti@unipg.it (R.G.); luca.mechelli@unipg.it (L.M.); fabrizio.passamonti@unipg.it (F.P.); 2Istituto Zooprofilattico Sperimentale del Piemonte, Liguria e Valle d’Aosta, National Reference Center of Veterinary and comparative Oncology (CEROVEC), 16129 Genova, Italy; mariaines.crescio@izsto.it (M.I.C.); laura.migone@izsto.it (L.F.M.); katia.varello@izsto.it (K.V.); paola.modesto@izsto.it (P.M.); elisabetta.razzuoli@izsto.it (E.R.); 3Dipartimento di Scienze Veterinarie, 10095 Grugliasco, Italy; tiziana.nervo@unito.it (T.N.); gianguido.donato@unito.it (G.G.D.); 4Dipartimento per la Innovazione nei sistemi biologici, agroalimentari e forestali (DIBAF), Università degli Studi della Tuscia, 01100 Viterbo, Italy; daniele.pietrucci@unitus.it (D.P.); gchillemi@unitus.it (G.C.); 5Dipartimento di Scienze Agrarie e Forestali (DAFNE), Università degli Studi della Tuscia, 01100 Viterbo, Italy; silvia.turco@unitus.it; 6Dipartimento di Statistica Economica, Università degli Studi Niccolò Cusano, 00166 Roma, Italy; carlo.drago@unicusano.it; 7Azienda USL Toscana Nord-Ovest UOC Ostetricia e Ginecologia, Nuovo Ospedale Apuane, 54100 Massa, Italy; alessandro.ghelardi@uslnordovest.toscana.it

**Keywords:** EcPV2, virus detection, gene expression, Italy, horse, subfertility

## Abstract

Equine Papillomavirus 2 (EcPV2) is responsible for squamous cell carcinomas (eSCCs) of external genitalia of both male and female horses. However, few studies report the EcPV2 prevalence among healthy horses. Currently, the lack of these data does not permit identifying at-risk populations and, thus, developing screening protocols aimed at the early detection of the infection, as for humans. The aim of our study was to estimate the genoprevalence of EcPV2 in clinically healthy horses in Italy and to evaluate their innate immune response. For this purpose, penile and vulvar swabs of 234 healthy horses were collected through sampling with sterile cytobrushes. Nucleic acids were isolated and EcPV2-*L1* presence (DNA) and gene expression (RNA) were checked by RT-qPCR. Our results showed EcPV2-*L1* DNA presence in 30.3% of the samples and *L1* expression in 48% of the positive samples. No statistically significant differences were found in genoprevalence in relation to sex, age, and origin, while, concerning breeds, the Thoroughbred had the highest risk of infection. Concerning specifically the mares, 40.2% of them resulted in being positive for EcPV2; our findings show a major positivity in pluriparous (*p* = 0.0111) and mares subjected to natural reproduction (*p* = 0.0037). Moreover, samples expressing *L1* showed an increased expression of *IL1B* (*p* = 0.0139) and *IL12p40* (*p* = 0.0133) and a decreased expression of *RANKL (p =* 0.0229) and *TGFB (p =* 0.0177). This finding suggests the presence of an effective immune response, which could explain the low incidence of SCCs in positive horses, despite a high EcPV2 genoprevalence (30%).

## 1. Introduction

Papillomaviruses (PVs) are small, 55 nm in diameter, DNA viruses characterized by a ds circular genome. PV genes are grouped into early (E) and late (L) based on their expression phase during the course of the infection. Their expression is regulated by a long upstream regulatory region (URR), which dictates tissue tropism and host range as well. Among the PV genes, the late L1 is the most conserved, and its sequence is widely used for PV detection and identification by sequencing.

PVs are characterized by tropism for cutaneous and mucosal keratinocytes and infect the basal layer through micro-abrasions [1]. After infection, the viral genome can be integrated into the host DNA or maintained as multiple episomes that replicate concomitantly with the host cells. Although the biological mechanisms related to tumor development and progression are unclear [2], viral integration has been associated with disease severity. In fact, it is known that the integration of viral DNA into the host genome causes the loss or disruption of the early E2 gene. E2 loss causes dysregulation of the E6 and E7 oncogenes, determining their increased expression and the consequent enhanced production of related oncoproteins [3,4].

PVs can cause both benign and malignant lesions in many species [1,5,6,7]. In humans, HPV infection is associated with cervical, anogenital, and head and neck squamous cell carcinomas (SCCs). Besides, there is growing evidence that PV infections may also play a role in cancer development in horses. In particular, EcPV2 has been detected in gastric, penile, vulvar, clitoral, and oro-pharyngeal papillomas, as well as in both in situ and invasive equine SCCs (eSCCs) [7,8,9]. Many studies reported the presence of viral DNA and mRNA in these lesions, suggesting EcPV2 as the etiologic agent [7,9,10,11]. Today, it is generally accepted that this virus plays an active role in the development of these tumors. However, little is known regarding the mechanism of oncogenicity, the transmission route (s), the existence of a carrier state, or the impact of PV status on prognosis or therapeutic protocols.

In humans, subclinical HPV infection is more frequent than the clinical disease, with many infections being cleared and few becoming persistent. In the horse, we can speculate that asymptomatic and transient EcPV2 infection is common; however, to date, only seven studies [11,12,13,14,15,16,17] have attempted to measure the prevalence of EcPV2 exposure or infection within the apparently healthy horse population, with non-Italian horses. Filling this void could allow identifying the at-risk populations and developing screening protocols for early detection and preventive measures’ application, as for humans.

Owing to the above, in this research, we evaluated the prevalence of EcPV2 DNA in the genitals of healthy horses resident in Italy. Moreover, we selected breeding stallions and broodmares and evaluated the genoprevalence in the mares.

## 2. Materials and Methods

### 2.1. Selection of Horses for EcPV2 Detection

Horses admitted to the Veterinary Teaching Hospitals of Perugia (OVUD) and Turin (OVU) from July 2020 to June 2021, for causes not related to pathologies of the genital system, were selected for genital swab collection. Inclusion criteria were the lack of any kind of neoplasia or PV-associated disease history. No restrictions were placed on age, breed, or sex. Cytobrush (Deltalab SLU, Barcelona, Spain) sampling was carried out with sterile cytobrushes, mildly rubbing the glans mucosa close to the fossa glandis for stallions and geldings and the vulvar mucosa in mares. The brushes were moisturized in 800 µL of DNA/RNA Shield Stabilization Solution (Zymo Research, Irvine, CA, USA), then stored at −20 °C until DNA and RNA extractions.

### 2.2. EcPV2 DNA Real-Time PCR

Total DNA of each sample was extracted from 200 µL of DNA/RNA Shield (Zymo Research, Irvine, CA, USA) using the QIAamp DNA Mini Kit (Qiagen, Milan, Italy), according to the manufacturer’s instructions. Samples were resuspended in 100 µL of elution buffer (Qiagen, Milan, Italy), and the DNA concentration was measured using a Qubit fluorimeter (Thermo Fisher Scientific, Waltham, MA, USA). EcPV2 detection was assessed in 100 ng of DNA sample. In particular, *L1*, *E2*, *E6*, and *E7* amplification by the real-time PCR assay was performed applying previously described protocols [10,18,19] and using the amplification of the equine beta-2-microglobulin (*B2M*) gene [10,20] as the control for DNA integrity (primer and probes sequences are reported in Table 1). Moreover, internal controls (block blanks, extraction blanks, and positive controls) were used for each analytical session. Samples were considered positive for EcPV2 if a Cq < 38 for the *L1* gene was observed. Moreover, some samples positive for EcPV2 were randomly selected and Sanger sequencing for L1 was performed. After amplification, purification, and sequencing, capillary electrophoresis was performed (ABI PRISM 310 genetic analyzer; Thermo Fisher Scientific, Waltham, MA, USA). A consensus sequence of the L1 gene was determined by the alignment of the forward and reverse strand using SeqMan (Lasergene package, DNASTAR Inc., Madison, WI, USA) for each sample tested.

### 2.3. NGS and Bioinformatics Analysis

Among all the samples included in the study, five horse samples from different Italian regions (one from Latium, two from Piedmont, one from Umbria, and one from Emilia Romagna) that tested positive by the real-time PCR were chosen for downstream NGS analysis. Total DNA was extracted and sequenced at the Institute of Applied Genomics (IGA, Udine, Italy) with the genome sequencer Illumina NovaSeq 6000, using paired-end sequencing of 150 bp. Raw reads’ quality was evaluated using FastQC [21]; NovaSeq 6000 adapters were trimmed using Trimmomatic v0.39 [22] using the following parameters: LEADING:15 TRAILING:15 SLIDINGWINDOW:4:15 MINLEN:111 HEADCROP:11. Trimmed reads were filtered through the Equus caballus reference genome (EquCab2.0, GCF_000002305.2) using BWA v 0.7.12 and samtools v1.13 [23,24]. The filtered reads were then assembled with SPAdes v3.15.4 [25] using default parameters. To check for the presence of PV in the assembled contigs, an in-house blast database was built using twenty-one complete genomes of different types of equine PVs retrieved from the National Center for Biotechnology Information (NCBI) database (Appendix A). Thus, the assembled contigs were blasted towards the in-house database using the BLASTn algorithm with default parameters, and those contigs that gave positive hits were manually processed in order to reconstruct the complete genome. The reconstructed genomes were further verified by aligning the filtered reads using BWA; the alignment was visualized with the Integrative Genomics Viewer (IGV) [26], and the breadth and depth coverage was calculated using samtools. The Genome Annotation Transfer Utility (GATU) [27] was used to identify the coding sequences of the reconstructed viral genomes. For the phylogenetic analysis, the capsid protein L1 sequences from samples 7, 10, and 12 were aligned to the L1 sequences of the reference type 2 genomes in Appendix A and to two more Italian L1 sequences (accession numbers MT063186.1 and MT063185.1) using MUSCLE [28]. The L1 sequence of the isolate BTU-1 of EcPV 4 was used as an outgroup (JF939718.1). The alignment was used as the input to build an ML tree with RAxML-HPC, using the GTRCATI algorithm as the substitution model and 1000 bootstraps [29].

### 2.4. RNA Extraction and EcPV2 Gene Expression

For each sample, 800 µL of DNA/RNA Shield (Zymo Research, Irvine, CA, USA) was used for total RNA extraction through the RNeasy Mini Kit (Qiagen, Milan, Italy), following the manufacturer’s instructions. Samples were resuspended in 50 µL of ultrapure RNAse-free H2O (Qiagen, Milan, Italy), and the RNA concentration was measured using a Qubit fluorimeter (Thermo Fisher Scientific, Waltham, MA, USA). Samples with at least a 25 ng/µL RNA concentration were used for gene expression analyses. The *E6* and *L1* EcPV2 genes were tested for their expression using specific primer sets and probes (Table 1) [12,13]. For reverse transcription (RT), 250 ng of RNA was used through the SuperScript™ IV VILO™ Master Mix (Thermo Fisher Scientific, Waltham, MA, USA). Then, 5 μL of 1:10 diluted cDNA was added to 20 μL of 1×iTaq Universal Probs Supermix (Bio-Rad, Irvine, CA, USA), as previously described [14]. Internal controls (block blanks, extraction blanks, and positive controls) were applied for each analytical session, and RNA was directly used to verify the lack of DNA amplification. A threshold of Cq < 38 for *L1* was used as the positivity assessment of viral gene expression.

### 2.5. Host Gene Expression Study

The host gene expression modulation was evaluated comparing EcPV2-positive and -negative horses. In particular, horses were divided into 3 groups: controls (C), EcPV2-L1p (samples with *L1* expression), and EcPV2-L1n (samples with *L1* DNA, but no expression). Gene expression of the receptor activator of nuclear factor-Kappa B ligand (*RANKL*), interleukin-1 Beta (*IL1B*), the IL-8 gene (*CXCL8*), *IL12p35*, *IL12p40*, and transforming growth factor beta (*TGFB*) was checked. *B2M* was used as a reference gene.

The primer set is reported in Table 2. RT-qPCR was performed using SsoFast™ EvaGreen^®^ Supermix (Bio-Rad, Hercules, CA, USA), as previously described [10]. Each sample was tested in triplicate, and the relative expression was calculated through the ΔΔCq method.

### 2.6. Statistical Analysis

Horses were categorized by age as “young” (8 years and younger) or “old” (9 years and older) using submission information. Moreover, breed and origin information were extracted from medical records. Microsoft Excel (2016) software was used for descriptive statistical analysis such as the mean ± 1 standard deviation and median calculations of age, male and female, and proportion of positivity. Moreover, STATA16.1 (StataCorp, college Station, Texas, USA) software was used to fit a logistic regression model assessing the association, expressed as the odds ratio (OR) between the positivity or negativity for EcPV2-L1 (dependent variable), 4 classes of age (<6yy, 6-<9yy, 9-<13yy, ≥13yy), and 2 of breed (English Thoroughbred vs. the others) (independent variable). The analysis was then restricted to the mares: a logistic regression was fit to assess the OR between the positivity or negativity for EcPV2-*L1* (dependent variable) and artificial insemination (AI) vs. natural service (NS) and to be pluriparous vs. maidens, taking into account breed and age (independent variables). Finally, a third logistic regression model was fit to assess the OR between being pregnant vs. empty (dependent variable) and the positivity or negativity for EcPV2-*L1* and being barren (independent variables).

For the gene expression analyses, the significant differences were evaluated through ANOVA followed by Dunnett’s multiple comparison test or a Kruskal–Wallis test followed by Dunn’s multiple comparison test. The significance threshold was set at *p* < 0.05.

## 3. Results

### 3.1. Sampled Population

A total of 234 horses (200 females and 34 males) were sampled in this study. The age range was 3 months–24 years, with a mean of 9 ± 5 years and a median of 9 years. Overall, 43.6% of the sampled animals was <8 years old (young) and 54.4% ≥ 8 years old (adults). Moreover, according to the 4 categories of age division, 58 animals were <6yy (very young), 45 of 6-<9yy (young animals), 68 of 9-<13yy (adult animals), and 62 of age ≥13yy (elderly animals). The sampled animals belonged to different breeds: 86 Thoroughbred, 57 Italian Standardbred, 26 Italian Saddle, 18 Arabian Horse, 6 Quarter Horse, 6 Shire, 5 Hannover, 5 Belgian, 5 Maremmano (Appendix A). Moreover, the breed was not known for 20 horses (Appendix A).

Sampled animals came from various Italian regions, as shown in Appendix A. In particular, 93 were from Piedmont, 47 from Umbria, 25 from Lazio, 22 from Tuscany, 11 from Marche, 10 from Emilia-Romagna, 4 from Lombardy, 4 from Sardinia, 3 from Campania, and 15 from other areas (Appendix A).

### 3.2. Detection of EcPV2 DNA

Overall, 30.3% (71 out of 234) of the tested animals resulted in being positive for EcPV2-*L1* DNA (Appendix A); of these, 14% (10 out of 71) were male and 86% (61 out of 71) female. Sanger analysis confirmed EcPV2 in all samples tested (Appendix A)

All samples positive for EcPV2-*L1* resulted in being positive for EcPV2-*E6* and *E7*. Only 76% (54 out 71) of the animals were positive for EcPV2-*E2* (Appendix A). The statistical analysis did not show significant differences in the genoprevalence linked to sex (*p* > 0.05) or origin (*p* > 0.05). In Figure 1, the sample consistencies per breed are reported: 50% of the Shire horses and 44.2% of the Thoroughbreds were positive for EcPV2. Moreover, EcPV2 was found in 33.3% of the Quarter Horse, 23.1% of the Italian Saddle, 26.3% of the Italian Standardbred, and 20% of the Hannover, Belgian, and Maremmano breeds. Only 5.6% of the Arabian Thoroughbred was infected.

The logistic regression model showed that Thoroughbreds had an OR of 3.3 (95% CI 1.8–6.2) to be positive for EcPV2 with respect to other breeds used as the comparison variable.

Regarding the age (Figure 2), a higher percentage of positive subjects in Groups 2 (52.3%), 3 (30.9%), and 4 (25.8%) compared to Group 1 (19%) was observed. In youngsters (6-<9yy) and adults (9-<13yy), significant associations with positivity for EcPV2 with OR 4.5 (CI95% 1.7–11.6) and 2.5 (CI95% 1.01–6.1) when compared with very young subjects (<6yy old) were found. No differences between very young subjects (<6yy) and old subjects (13 yy and more) were observed.

### 3.3. Detection of EcPV2 DNA in Mares

One-hundred twenty-nine broodmares, selected from the total sampled horses, permanently or temporarily resident in the province of Turin, were analyzed to specifically evaluate the EcPV2 prevalence in breeding horses (Table 3). Mares belonged to different breeds: 68 Thoroughbred, 55 Italian Standardbred, 5 Shire, and one Italian Saddle.

Mares’ age ranged between 3 and 21 years, with an average of 10 ± 4 years and a median of 5 years. Thoroughbred and Shire were subjected to natural service and Italian Standardbred and Italian Saddle to artificial insemination. Among the sampled mares, 25 were primiparous (19%) and 104 pluriparous (81%). During our study, following the breeding season (2020–2021), 78 foals were born; 51 of the followed mares did not become pregnant or did not carry the pregnancy to term. Of the mares, 40.2% (51 out of 129) resulted in being positive for EcPV2: 25.5% of the Italian Standardbred, 48.5% of the Thoroughbred, and 60% of the Shire (Table 4). Our data show a significant difference (*p* = 0.0111) in EcPV2 genoprevalence between maiden and pluriparous; indeed, 16% of primiparous (4 out of 25) and 45% of pluriparous (47 out of 104) were positive for EcPV2 (Table 4). Concerning the pregnancy, 12 mares were excluded from this analysis because they died before the end of pregnancy or because they were not submitted to insemination. Moreover, 33.3% of the mares that had a foal were positive for EcPV2, while 53.8% of mares that did not have a foal resulted in being positive (Table 4). Thus, a significant difference (*p* = 0.0452) in EcPV2 genoprevalence between mares with or without a foal was observed. Additionally, a significant difference (*p* = 0.0037) between NS and AI (Table 4) was found.

No association was found between the positivity or negativity for EcPV2-*L1* and artificial insemination vs. natural service and to be pluriparous vs. maiden, taking into account breed and age. Instead, being pregnant vs. empty was associated with both being barren (OR = 17.9, CI95% 6.0–53) and positive for EcPV2-*L1* (OR = 2.7, CI95% 1.04–7.3).

### 3.4. NGS and Bioinformatics Analysis

The total number of raw, trimmed, and filtered reads from the five selected samples is shown in Appendix A. The contigs obtained from the assembled filtered reads were blasted towards the in-house papillomavirus database, and all the samples showed contigs related to EcPV2 (Appendix A). In particular, samples ID2396_7-7, ID2396_10-10, and ID2396_12-12 showed one single contig (NODE_35, NODE_66, and NODE_3, respectively), of which the length already represented the complete EcPV2 genome, with an overall 100% coverage. Sample ID2396_4-4 showed several shorter contigs related to EcPV2, and its consensus sequence was retrieved through samtools and bcftools by mapping the filtered reads towards a reference EcPV2 sequence (LC612601.1), showing a genome coverage of 93.36%. For sample ID2396_1-1, instead, after mapping the filtered reads towards the reference, the genome coverage was only 70.46%. Nonetheless, to further verify their robustness, the filtered raw reads of samples ID2396_7-7, ID2396_10-10, and ID2396_12-12 were also aligned on the genome sequences reconstructed from the contigs and the alignment visualized with IGV (Appendix A). The phylogenetic analysis based on the alignment of the L1 gene sequence with the fully reconstructed sequences showed that the reconstructed 7, 10, and 12 samples clustered apart from the other type 2 isolates (Figure 3).

### 3.5. Detection of Gene Expression

Concerning the gene expression, a downregulation of *TGFB* (*p* = 0.0177) and *RANKL* (*p* = 0.0229) in animals EcPV2-L1p with respect to control animals was observed, while *IL1B* (*p* = 0.0139) and *CXCL8* (*p* = 0.0426) were upregulated. *IL12p35* was not modulated, and the expression of *IL12p40* was increased in animals EcPV2-L1n (*p* = 0.0133) with respect to the controls (Figure 4).

Appendix A shows the data related to the presence of EcPV2 in terms of DNA and the expression of the *L1* and *E6* genes. The obtained results showed that 52% of the animals (13 out of 25) were positive for both EcPV2 DNA and RNA, thus with the virus actively replicating, while in 48% of the animals (12 out of 25), viral replication was not detected. The data are expressed in Cq ± 1DS obtained from the average of three replicates.

## 4. Discussion

To date, only seven studies [11,12,13,14,15,16,17] have measured the prevalence of EcPV2 exposure or infection in healthy horses, and none of these included Italian horses. Therefore, in order to contribute to the discussion on EcPV2’s role as a risk factor in cancer development and its impact on fertility, in this study, the frequency of EcPV2 infection in asymptomatic horses was determined by analyzing genital swab samples. Moreover, to evaluate innate immune response, we assessed the gene expression of some important genes; indeed, it is known that inflammation plays a pivotal role in cancer development and progression [30,31].

Our study shows that EcPV2 infection occurs among healthy horses with an overall prevalence rate of 30.3%, which is very similar to the value reported by Grenwood and co-workers [17], especially for penile swabs (30%), but higher if compared to other previous studies where the reported genoprevalence ranged between 5 and 10% [7,11,12]. Regarding vulvar and vaginal swabs of healthy mares, the EcPV2 positivity in our study (29%) was higher than that reported in previous papers [12,13,15,32], where it ranged from 0 to 24%. These differences could also be attributed to the sampling methods, since our samples were collected through sterile cytobrushes rather than regular swabs. Furthermore, it is reasonable to hypothesize that, in asymptomatic horses, the number of viral copies is very low and limited to the deeper layers, as for HPV infection [33]. These hypotheses could partly justify the different results reported in various papers.

All samples positive for EcPV2-*L1* resulted in being positive for EcPV2-*E6* and *E7*, while only 76% (54 out 71) of them were positive for EcPV2-*E2* (Appendix A). Nonetheless, the EcPV2 presence was further confirmed through NGS analysis, and the complete viral genome was reconstructed from three different samples deposited under accession numbers ON942231, ON942232, and ON942233. Concerning positivity for E2, which was found only for 76% of the infected animals, it is important to remember that the viral genome can be integrated into the host DNA or maintained as multiple episomes that replicate concomitantly with the host cells. Moreover, it is known that the integration of viral DNA into the host genome causes the loss or disruption of the early E2 gene [11,12,13,14,15,16,17]. This could explain the absence of this gene for a part of the samples. In this study, EcPV2 genoprevalence was similar in horses of different sex, age (<8y and >8y), and origins, while, concerning the breeds, Arabian had the lowest and Thoroughbred the highest EcPV2 genoprevalence with respect to the other investigated breeds. This is an interesting finding, considering the evidence for horses’ genetic susceptibility to sarcoids [34,35]. In particular, it is known that the polymorphisms of the major histocompatibility complex (MHC) region are crucial in papillomavirus-related cancer progression. Indeed, Staiger and co-workers [35] identified a variation on the *DQA1* gene intron, in the MHC class II region on the Eca 20 chromosome, confirming the MHC variability association with sarcoid susceptibility. The difference in EcPV2 genoprevalence between breeds could be therefore due to genetic differences of the MHC, although further studies are needed to confirm this hypothesis. This relationship has also been demonstrated in human cervical carcinoma [36]. If confirmed, the link between MHC and PV-induced tumors in humans and horses would suggest a common pathogenic mechanism and the possibility to use the horse as a model in comparative oncology [10,18,37].

Looking at the stratified results with respect to age and breed, four classes of age (<6yy, 6-<9yy, 9-<13yy, ≥13yy) and two of breed (Thoroughbred vs. the others) were identified. Our data outline a major risk of infection in animals of Groups 2 (6-<9yy) and 3 (9-<13yy): analogous to humans, also in horses, the major risk of infection occurs at the beginning of the fertile age and, in particular, during the reproductive period [38]. Moreover, in our study, EcPV2 was also detected in a 6-month-old foal; this is not surprising given the recent documentation of EcPV2 vertical transmission, also reported for HPV and BPV [39,40]. Interestingly, in our study, mares subjected to natural mating resulted in having a major risk of being positive for EcPV2 with respect to mares undergoing artificial insemination, probably due to possible micro trauma, which induces mucosal abrasions with a greater probability of virus transmission.

However, it should be considered that all mares with natural mating were Thoroughbred, and therefore, a breed predisposition regardless of the mating type may also exist. Moreover, it is relevant to outline that maidens presented a lower genoprevalence than pluriparous mares, suggesting sexual transmission, as previously speculated both in humans and horses [17,41].

Only 50% of animals were positive for EcPV2-L1 gene expression; these differences could be due to low viral loads, as suspected in asymptomatic infections. Indeed, all animals in the current study were healthy. Another cause could be the sampling method: in fact, by cytobrush, only the superficial cells of the epithelium are recovered, and in these cells, the L1 and L2 genes are expressed, whereas E6 and E7 mRNA strongly decrease their expression [42,43,44].

Moreover, we decided to test host genes such as IL1B, CXCL8, TGFB, IL12p35, IL12p40, and RANKL because they are involved in immune response against PVs [37,45,46,47,48]. RANKL/RANK signaling is pivotal for bone homeostasis; this pathway has a functional role in normal tissues, and it is expressed in many epithelial and stromal structures. It is known that RANKL modulates immune response inducing dendritic cell (DC) survival and lymphocyte differentiation and activation. Moreover, it can be detected in tumor-infiltrating lymphocytes (TILs), B cells, immature DCs, monocytes, and macrophages. In many cancers, it plays a pivotal role in creating an immune microenvironment that promotes tumor progression. The RANK signal can drive epithelial to mesenchymal transition (EMT), promote osteomimicry, induce stem cell phenotypes, and determine metastasis [49]. These activities were shown both in horses and humans [10,18,50]. The TGFB consists of three isoforms secreted in a latent form and activated via various mechanisms. TGFB is a major EMT regulator and also an important mediator of the stromal environment changes that are essential in tumor progression [51]. In addition to driving tumor cell migration and metastasis, *TGFB* plays a key role in promoting human papillomavirus infection by weakening the host immune defense.

In our results, downregulation of *RANKL* and *TGFB,* two important cytokines involved in horse and human genital SCC development and progression [10,18], were observed in animals with productive infection (EcPV2-*L1* expressed). Thus, such downregulation could indicate a good immune response against viruses [51]. At the same time, the upregulation of *IL1B* and *CXCL8,* two pro-inflammatory cytokines involved in the innate immune response during the initial phases of viral infection, which could induce a good pro-inflammatory response to PVs, determining the clearance of the viral infection [52,53], was observed. These data could explain the high prevalence of EcPV2 infection and low incidence of genital SCC in horses and confirm the horse as a model in comparative oncology. Indeed, also in humans, 90% of HPV infections are cleared within 2 years following the success of immune response [53]. Our data could suggest inflammation in the genital tract, which could be related to viral infection; indeed, pro-inflammatory gene expression was observed only in horses that tested positive for L1 gene expression.

## 5. Conclusions

This paper provides for the first time the genoprevalence of EcPV2 in Italian horses. Our results suggest that in horses, as in humans, many infections are asymptomatic and resolve spontaneously; Thoroughbreds are more susceptible to the infection; in horses, as in humans, sexual transmission exists. Moreover, we also found a positive association between being barren and positive for EcPV2-L1, and although the number of samples, together with the molecular and morphological investigations are not sufficient to make speculations about it, we believe that these are interesting data to investigate further, given that in humans, the infection is associated with low fertility.

## Figures and Tables

**Figure 1 viruses-14-01696-f001:**
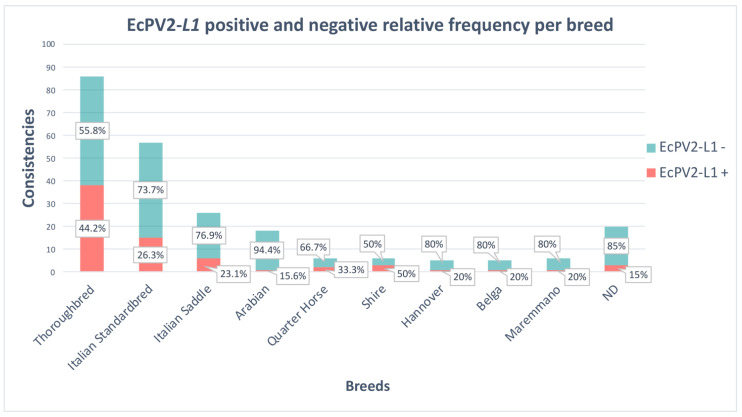
EcPV2 genoprevalence in the different breeds. Bars represent the number of sampled animals for each breed with the orange part depicting the positivity for EcPV2-L1 and the green part the negativity for the virus (with relative percentages).

**Figure 2 viruses-14-01696-f002:**
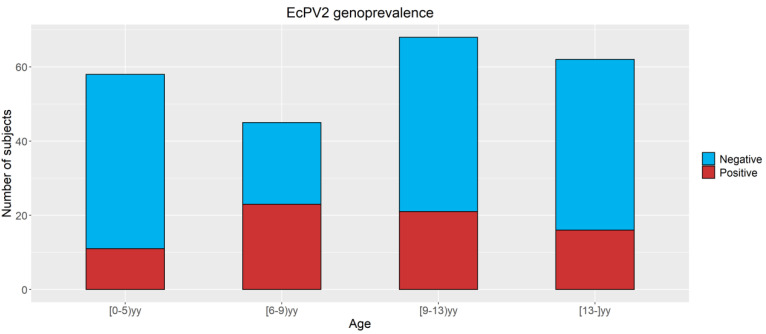
EcPV2 genoprevalence based on age classes. Positive (red part) and negative (blue part) number of subjects for EcPV2-L1.

**Figure 3 viruses-14-01696-f003:**
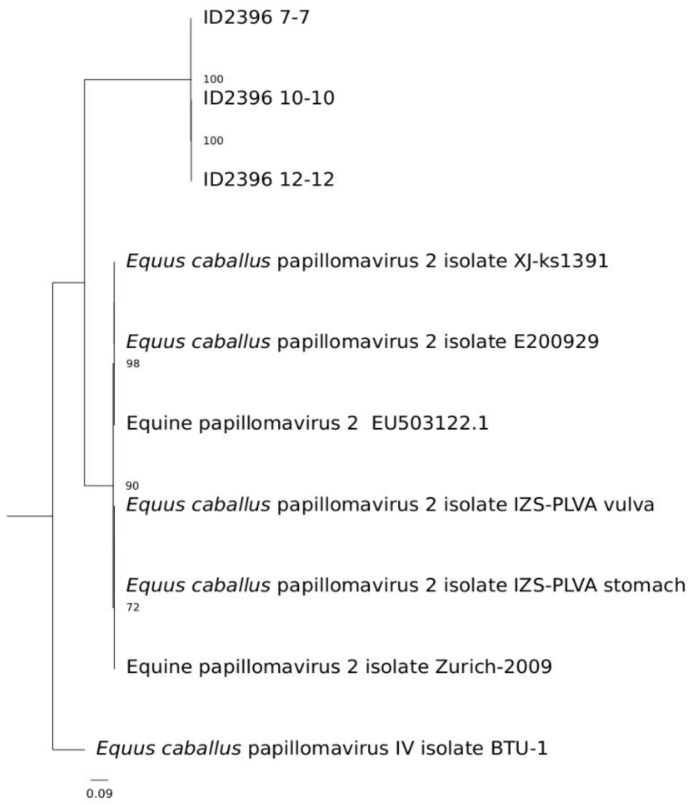
Phylogenetic analysis of the L1 gene sequence from the three reconstructed samples (ID2396 7-7-, 10-10, and 12-12) and other EcPV2 isolates available in the NCBI database. EcPV4 was used as the outgroup, and the bootstrap value indicating clustering robustness is indicated.

**Figure 4 viruses-14-01696-f004:**
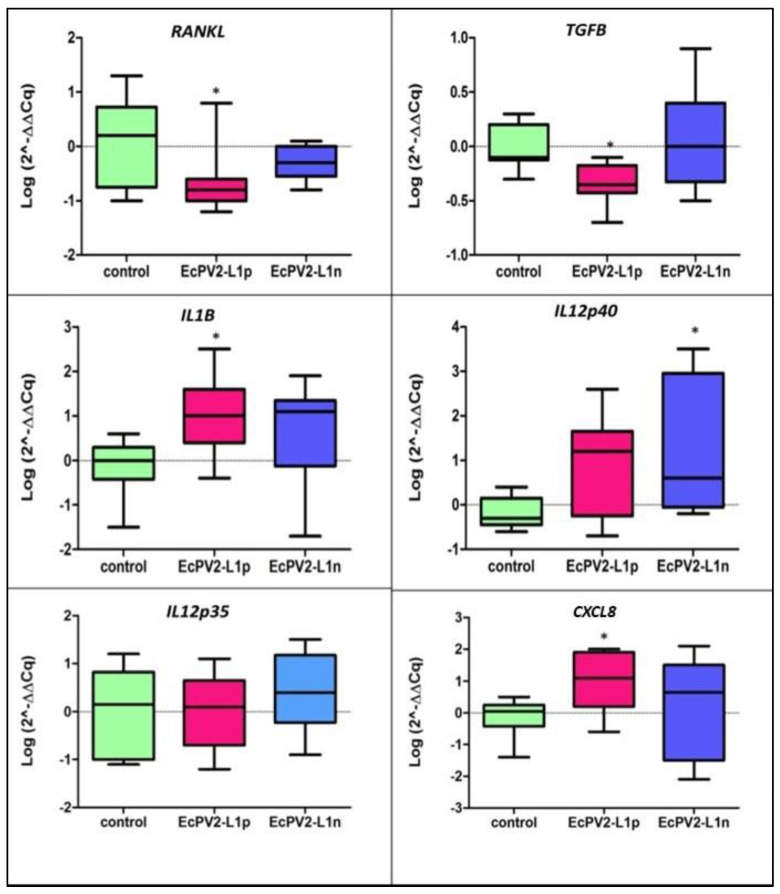
Horse gene expression results. Data are represented as box-and-whisker plots displaying median and interquartile ranges (boxes) and minimum and maximum values (whiskers). Differences (control vs. EcPV2-L1p; control vs. EcPV2-L1n) were evaluated through ANOVA followed by Dunnett’s multiple comparison test or a Kruskal–Wallis test followed by Dunn’s multiple comparison test; * *p* < 0.05.

**Table 1 viruses-14-01696-t001:** Primer sets and probes for DNA and gene expression detection.

Gene	Sequences	Amplicon Length	Accession Number
EcPV2-*L1*	F-5′-TTGTCCAGGAGAGGGGTTAG-3′ R-5′-TGCCTTCCTTTTCTTGGTGG-3′	80	NC_012123.1:5673-7172
p-EcPV2-*L1*	FAM-CGTCCAGCACCTTCGACCACCA-TAMRA		
EcPV2-*E6*	F-5′-CGTTGGCCTTCTTTGCATCT-3′ R-5′-AGGTTCAGGTCTGCTGTGTT-3′	80	NC_012123.1:5-622
p-EcPV2-*E6*	FAM- CCGTGTGGCTATGCTGATGACATTTGG-TAMRA		
EcPV2-*E7*	F-5′-CTCTGAGCAGCATCACCCTT-3′ R-5′- TCTTCCTCGTCTTCTGTGTCC-3′	70	NC_012123.1:624-959
p-EcPV2-*E7*	FAM- AGAGCGCTCCCCCTCAGTCA-TAMRA		
EcPV2-*E2*	F-5′-AAAAGGGAGGGTACGTTGTC-3′ R-5′-CCTGGTAGTAGACATGCTGC-3′	90	NC_012123.1:2767-4017
p-EcPV2-*E2*	FAM- GCCAAGACAGCCACGACGCCAT-TAMRA		
*B2M*	F-5′-GGCTACTCTCCCTGACTGG-3′ R-5′- TCAATCTCAGGCGGATGGAA-3′	135	NM_001082502.3
p-*B2M*	FAM-ACTCACGTCACCCAGCAGAGA-TAMRA		

**Table 2 viruses-14-01696-t002:** Primer sets for gene expression detection.

Gene	Primer Pairs Sequences	Amplicon Length	Accession Number
*RANKL*	F-5′-AGCCTGACACTCAACCTTTTG-3′ R-5′-CCAGGAAGACAGACTCACTTTG-3′	86	XM_014732051.2
*IL12p35*	F-5′-CTGAGGACCGTCAGCAACAC-3′ R-5′-GTTCGGGGCGAGTTCCAG-3′	147	NM_001082511.2
*IL12p40*	F-5′-GATCGTGGTGGATGCTGTTC-3′ R-5′-TCCACCTGCCGAGAATTCTT-3′	132	NM_001082516.1
*CXCL8*	F-5′-CTGGCTGTGGCTCTCTTG-3′ R-5′-CAGTTTGGGATTGAAAGGTTTG-3′	133	NM_001083951.2
*IL1B*	F-5′-TGATGCAGCTGTGCATTCAGT-3′ R-5′-GCACAAAGCTCATGCAGAACA-3′	146	NM_001082526.1
*TGFB*	F-5′-CGGAATGGCTGTCCTTTGATG-3′ F-5′-CCCACGCGGAGTGTGTTAT-3′	127	NM_001081849.1

**Table 3 viruses-14-01696-t003:** EcPV2 genoprevalence in mares divided by breeds.

	Italian Standardbred	Thoroughbred	Shire	Italian Saddle
Positive	14	33	3	0
Negative	41	35	2	1
Total	55	68	5	1

**Table 4 viruses-14-01696-t004:** EcPV2 genoprevalence of the mares.

	Positive	Negative	Total
Maiden	4	21	25
Pluriparous	47	57	104
Total	51	78	129
*p*-value	0.0111		
No foal	21	18	39
Foal	26	52	78
Total	47	70	117
*p*-value	0.0452		
Natural service	34	31	65
Artificial insemination	16	48	64
Total	50	79	129
*p*-value	0.0037		

## Data Availability

Not applicable.

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
