# Peer review of "Detection of Equus Caballus Papillomavirus Type-2 in Asymptomatic Italian Horses"

_viruses, 2022, doi:10.3390/v14081696_

Round 1

Reviewer 1 Report

I read the paper by Cappelli et al. This paper in its present form does not meet any requirements for publication in the "Viruses". It is a potential, intriguing paper but it is written in a not-well organized and confusing manner. They have to allow an Editing Service from any Publishers to improve English language;

just an example. line 22 change... responsible of... in responsible for...please;

line 42 change, please, "single" in "ds" (see Ramsauer et al., Viruses 2021, 13(7): 1404);

the authors should describe in details both DNA and cDNA of L1, E6, and E7. They should perfom sequencing of these genes and show the molecular finding.

Furthermore, it is non enough to state....all samples positive to EcPV2-L1 resulted positive for E6 and E7 (line 182). It is not correct to state it without showing any molecular findings They should show molecular findings to do this statement;

The authors should present their results by FIGURES. 

Lines 127-129: the authors should explain why they are investigating RANKL etc.... They suggested that these genes are involved in immune responses. In general, this is true but the authors should show that PV infection is correlated with immune responses.  

Line 302: Animals harboring expression of L1 have not ABSOLUTELY an active form of PV infection as sggested by authors. It can mean that we have a productive infection. It is necessary to investigate E6 and E/ to show that the infection is abortive and associated with tumors.

Reviewer 2 Report

Detection of Equus caballus Papillomavirus Type-2 in Asymptomatic Italian Horses: possible effect on fertility

by Katia Cappelli, Chiara Grazia De Ciucis, Samanta Mecocci, Tiziana Nervo, Maria Ines Crescio, Marco Pepe, Rodolfo Gialletti, Laura Federica Migone, Luca Mechelli, Fabrizio Passamonti, Carlo Drago, Gian Guido Donato, Katia Varello, Paola Modesto, Alessandro Ghelardi, and Elisabetta Razzuoli.

The manuscript describes evaluation of the genoprevalence of EcPV2 in 234 healthy Italian horses. Several factors were evaluated to lead to higher risk of EcPV2 infections. Furthermore, the authors estimated the viral gene expression of L1 and E6 as well as for a collection of host genes (RANKL, IL1B, IL8, IL12p35, IL12p40, TGFB) by means of RT-qPCR. The study is interesting and worth to publish. However, there are some issues which have to be addressed:

Major concerns:

- The B2M-primers and probe binding sites are separated by an exon-exon junction on the beta-2-microglobulin mRNA. Since the binding sites of primer and probes for detection of L1- and E6-mRNA are not separated by an exon-exon junction, a control reaction omitting the reverse transcriptase should be included to rule out DNA contamination of the RT-qPCR. 

- As indicated in the title of the manuscript, the authors presume a certain effect of EcPV2 infection on fertility. The p-values to support this hypothesis are (weakly) significant. The rational to choose the host genes for expression analysis is not clear. In particular, a connection to fertility is not given or explained nor discussed. 

- As the qPCR for detection of L1, E2, E6 and E7 target the same molecules (the viral genome) it is expected that these should result in a similar amplification (Table S1 and S2). If the efficiencies of the reactions are different the Cq-values might be different but showing the same trend. It is not expected that in some cases the E2 region of the genome is not amplified. A possible explanation would be the insertion of the genome and thereby the loss of some parts of the viral genome. However, this is not discussed. 

Minor concerns:

- Why a logistic regression model is applied instead of a simple Fisher or Chi2 test to calculate the odd rations and p-values?

- It should stated in the description of Table S1 that the values are based on qPCR to detect viral DNA. 

- A qPCR was performed for E2 and E7 (Table S1 and S2) however, the primers used for the reactions are not listed. 

- What are the criteria to choose the (some) samples listed in Table S1? Were all 71 DNA-positive samples tested for the viral gene expression?

- There is no graph for CXCL8 in Figure 1 although discussed in the text. 

- There are some minor mistakes in the text: [line 36] the 'p' for p-value is given as majuscule instead of minuscule. [line 74] 'This gap gives impossible to identify....' should be rephrased [line 180] 'The 30% (...) of animals were positive...' should be rephrased [line 319] '... immune response agonist viruses... ' might be replaced with 'against'.

Round 2

Reviewer 1 Report

Desèite the fact the authors improved their manuscript, it seems to me that  this paper does not correlate, in its present form, the presence of virus and the pathological conditions the authors assumed to be. The authors MUST SHOW gene expression of the virus. In particular, they have to show the expression of Early genes involved in abortive infection. The presence of papillomavirus in genital apparatus can not have any biological significance unless morphological and more appropriate molecular findings are shown. In other terms, the author should document a convincing study showing gene expression of virus. Alternatively, the authors could study morphological abnormalities related to PV- infections such as abnormalities involving axonemal morphology.
